# Using testing history to estimate HIV incidence in mothers living in resource-limited settings: Maximizing efficiency of a community health survey in Mozambique

Orvalho Augusto[1,2,3☯], Sheila Fernández-Luis[1,4☯]*, Laura Fuente-Soro[1,4], Tacilta Nhampossa[1,5], Elisa Lopez-Varela[1,4], Ariel Nhacolo[1], Edson Bernardo[1,6], Helga Guambe[7], Kwalila Tibana[7], Adelino Jose Chingore Juga[8], Jessica Greenberg Cowan[9], Marilena Urso[8], Denise Naniche[4]

1 Manhiça Health Research Centre (CISM), Maputo, Mozambique, 2 Faculty of Medicine, University Eduardo Mondlane, Maputo, Mozambique, 3 Department of Global Health, University of Washington, Seattle, Washington, United States of America, 4 ISGlobal, Hospital Clínic, Universitat de Barcelona, Barcelona, Spain, 5 Instituto Nacional de Saúde (INS), Maputo, Mozambique, 6 Manhiça District Health Services, Maputo Province, Mozambique, 7 Ministério da Saúde de Moçambique (MISAU), Maputo, Mozambique, 8 Division of Global HIV and Tuberculosis, U.S. Centers for Disease Control and Prevention, Maputo, Mozambique, 9 Maternal and Child Health Branch Chief, Mozambique, Division of Global HIV and TB, Centers for Disease Control and Prevention, Maputo, Mozambique

☯ These authors contributed equally to this work.
* Sheila.fernandez@isglobal.org

## Abstract

Obtaining rapid and accurate HIV incidence estimates is challenging because of the need for long-term follow-up for a large cohort. We estimated HIV incidence among women who recently delivered in southern Mozambique by leveraging data available in routine health cards. A cross-sectional household HIV-testing survey was conducted from October 2017 to April 2018 among mothers of children born in the previous four years in the Manhiça Health Demographic Surveillance System area. Randomly-selected mother-child pairs were invited to participate and asked to present documentation of their last HIV test result. HIV-testing was offered to mothers with no prior HIV-testing history, or with negative HIV results obtained over three months ago. HIV incidence was estimated as the number of mothers newly diagnosed with HIV per total person-years, among mothers with a prior documented HIV-negative test. Among 5000 mother-child pairs randomly selected, 3069 were interviewed, and 2221 reported a previous HIV-negative test. From this group, we included 1714 mothers who had taken a new HIV test during the survey. Most of mothers included (83.3%,1428/1714) had a previous documented HIV test result and date. Median time from last test to survey was 15.5 months (IQR:8.0–25.9). A total of 57 new HIV infections were detected over 2530.27 person-years of follow-up. The estimated HIV incidence was 2.25 (95% CI: 1.74–2.92) per 100 person-years. Estimating HIV incidence among women who recently delivered using a community HIV-focused survey coupled with previous HIV-testing history based on patients' clinical documents is an achievable strategy.

**Data Availability Statement:** There are ethical restrictions on sharing a de-identified data. Data

contain potentially sensitive information and national ethics committee does not authorize data sharing without a protocol request specifying the objectives and the researchers who will have access to the data. Data are available under request (contact via llorenc.quinto@isglobal.org) for researchers who meet the criteria for access to confidential data.

**Funding:** This evaluation was supported by the President's Emergency Plan for AIDS Relief (PEPFAR) through the U.S. Centers for Disease Control (CDC) under the terms of CoAg GH000479. The findings and conclusions in this report are those of the author(s) and do not necessarily represent the official position of the funding agencies. AJCJ, JGC and MU received salary from PEPFAR through the U.S. CDC. ELV received funding from the European Respiratory Society and the European Union's H2020 research and innovation program under Marie Sklodowska-Curie grant agreement no. 847462. The funders had no role in study design, data collection and analysis, decision to publish, or preparation of the manuscript.

**Competing interests:** The authors have declared that no competing interests exist.

## Introduction

HIV prevalence estimates are widely used to monitor the course of the HIV epidemic. However, longer survival of people living with HIV (PLHIV) resulting from antiretroviral treatment (ART) expansion makes it difficult to interpret changes in HIV prevalence over time. HIV incidence, which is the rate of new infections in a population over time, is less frequently reported despite being a more reliable indicator of transmission [1, 2].

Measuring population-based HIV incidence allows for monitoring epidemic dynamics, evaluating the effectiveness of HIV prevention and treatment programs, and identifying and targeting populations at risk [1, 3]. Nevertheless, there are formidable challenges to measuring community-level HIV incidence. Direct prospective measurement of incidence involves large sample sizes, longitudinal follow-up and regular retesting of HIV-negative individuals to identify incident cases [3]. Despite not necessarily reflecting the true rate of new infections, the number of newly HIV diagnosed individuals is often used as a proxy for incidence [4]. Indirect incidence estimates can be based on model-derived calculations from serial prevalence surveys, from recently exposed populations, or using assumptions about risk behaviour and HIV transmission [2]. Other incidence estimation methods involve using biomarkers for recent HIV infection and then using mathematical procedures to derive HIV incidence [2]. Whether through model-derived calculations or biomarker-based assessments, simpler, reliable and sustainable methods for HIV incidence estimation in countries with the highest burdens of HIV are needed. One strategy to obtain rapid and accurate estimates of HIV incidence in low-resource settings could be to use HIV-testing history from mothers' health cards combined with HIV test results obtained during periodic demographic and health surveys.

We estimated HIV incidence among women who had given birth in the previous four years in the Manhiça district in southern Mozambique, using previous HIV-testing history documented in routine maternal and child health cards during a community HIV-testing survey.

## Methods

### Ethics statement

This study was approved by the Mozambican National Bioethics Committee, the Institutional Review Boards at the Hospital Clinic of Barcelona (Spain), the CISM, and reviewed in accordance with the U.S. Centers for Disease Control and Prevention (CDC) human research protection procedures and determined to be research, but CDC investigators did not interact with human subjects or have access to identifiable data or specimens for research purposes. Written informed consent was obtained for all women before the interview.

### Study setting

The Manhiça Health Research Center (In Portuguese: *Centro de Investigação em Saúde da Manhiça* [CISM]) runs the Health and Demographic Surveillance System (HDSS), which documents vital events such as births, deaths, pregnancies, migrations causes of deaths using verbal autopsies, by a combination of visits to each household (twice a year), weekly visits to key informants and daily visits to health facilities [5, 6]. This surveillance covers about 45 000 households and 200 000 inhabitants, who are assigned a unique identification number through which all the events are linked to the person and household where he/she lives [5, 6].

Manhiça district's community HIV prevalence is almost 40% among adults, almost three times higher than the national adult HIV prevalence of 13.2% [7, 8]. Manhiça district has an antenatal care (ANC) coverage of 74% of women attending at least one ANC visit in the last pregnancy, of whom 78.8% receive HIV counseling and testing [9]. Quarterly HIV-testing is

recommended during pregnancy [10, 11]. Following national recommendations, pregnant women have a prenatal card documenting antenatal visits, including HIV-testing during pregnancy and delivery. After delivery, the mother receives a child's health card containing birth and postnatal data, and mother and child HIV-testing information.

## Study design

This analysis used data from a broader cross-sectional household study [12]. The cross-sectional household study used simple random sampling to identify 5000 children born alive within the four years before study implementation (September 1, 2013–October 31, 2017) from a list of mothers aged > 14 years and residing in the HDSS area. A household visit was conducted between October 2017 and April 2018, and consenting mothers/caregivers completed a study-specific questionnaire. For each participant, three household visit attempts were made before defining the status as absent. If the mother was absent, migrated, or had died, the child's main caregiver provided informed consent and completed the survey. During the survey, mother and child HIV status was determined through documentation, age-appropriate testing, laboratory confirmation, or verbal autopsy (for further information on broader cross-sectional household see Fuente-Soro et al. [12]).

For incidence estimation, we included those mothers interviewed in the cross-sectional household study with a prior negative HIV test within the last four years and who received a new test at survey. Mothers without a previous negative HIV test or without a HIV test during the survey and household visits were excluded from incidence analysis. Caregivers other than mothers were excluded from this analysis.

## Study procedures

Information regarding prior HIV-testing was requested from all participants during the HIV-testing survey. Participants were asked to self-report their HIV status and to present documentation of their last HIV tests, such as the pregnancy card, child health card, HIV care card, or HIV counseling and testing card. Following national testing recommendations, participants that, at the time of the study visit, did not know their HIV status or had tested HIV-negative more than three months before the study visit, were offered testing according to the national HIV-testing algorithm, which includes two serial HIV rapid diagnostic tests from different manufacturers (Determine and Unigold) [13–15]. For all HIV-positive participants, a dry blood spot (DBS) was collected on filter paper to perform a laboratory confirmation test (Bio-Rad Geenius HIV1/2 Confirmatory assay) [16]. The study HIV counselors provided pre- and post-test counseling, and all newly diagnosed women were referred to the nearest health unit for HIV care and treatment. The study counselors provided a referral guide and and recorded the data of the patient and of the nearest health unit or the one of preference in a study logbook, in order to facilite linkage to care.

## Data management

Data was collected on electronic questionnaires using the ODK 1.4 platform (https://opendatakit.org/) [17]. Automatic data entry checks and skip patterns were implemented to ensure data quality. Data was uploaded to the server daily. Data managers checked for form completeness, and incomplete data fields were corrected when possible. The questionnaires included demographic variables (HDSS identifier, date of birth, marital status, education level and occupation), HIV knowledge, self-awareness of HIV status, and the result of the current HIV test.

The data were exported and merged with the HDSS member status for analysis in Stata 15 (StataCorp. 2017. Stata: Release 15. Statistical Software. College Station, TX: StataCorp LLC).

## Statistical analysis

We estimated the number of HIV-positive results obtained during the home visit among mothers with a previous HIV-negative diagnosis. HIV incidence rate was computed as the quotient of total positive HIV tests and the total person-time contributed since the last negative test. For mothers with a previous negative test without a registered date, delivery date was used as the last test date, in accordance with the national testing recommendation at delivery. For women testing HIV-positive at the survey, the seroconversion date was established at a random-point date between the last negative result and the visit date, as recommended in the literature [18, 19]. Person-time contribution was computed from the last test date to the estimated seroconversion date. Poisson-distribution-based 95% confidence intervals (CI) were computed [20]. Two sensitivity analyses were conducted, one where the infection date was considered as the date of the survey (no imputation of the date of infection), and another including women removed from the main analysis due to a negative test within the previous three months.

Descriptive analysis was conducted. For continuous variables, means and standard deviation (SD), median and interquartile ranges (IQR) were used. For categorical variables, frequencies (counts and proportions) were employed. Comparisons between groups were made using Pearson chi-square or Fisher exact test and Kruskal Wallis tests, as applicable.

## Results

Among the 5000 mother-child pairs selected, 96.5% (4826/5000) were eligible for the main study and 63.6% of the eligible mothers (3069/4826) were found during the household visit and interviewed (Fig 1). The majority of interviewed mothers (99.5%, 3055/3069) reported a previous HIV test. A total of 26.9% (823/3055) of the mothers with a previous test had a self-reported or documented HIV-positive result, 72.7% (2221/3055) an HIV-negative result, 0.2% (7/3055) an indeterminate result and 0.1% (4/3055) did not remember their previous result.

Among the 2221 HIV-negative mothers, 77.2% (1714/2221) were included in the analysis and 22.8% (507/2221) were excluded mainly due to test refusal (29/507) or a documented negative test in the previous three months (478/507) which made them ineligible for a new test, according to national guidelines [15, 21, 22]. Additional information about the study sample is provided elsewhere [12].

A total of 1714 HIV-negative women receiving a new HIV test in the survey were included in this analysis.

### Clinical documentation showed at survey

A total of 93.0% (1594/1714) of mothers showed some sort of clinical documentation. The majority presented both a prenatal card and child's health card (Table 1). The child health card coverage was 72.2% (1237/1714). A total of 83.3% (1428/1714) of the mothers had a dated HIV test result registered in their clinical documentation. Among mothers with a dated HIV test result, median time from last test to survey was 15.5 (IQR:8.0–25.9) months.

### Sociodemographic characteristics of mother-child pairs at survey

The median age for mothers was 24.8 years (IQR: 20.7–31.5) (Table 2). Most were married/in a marital union (79.6%) and had only primary school education (71.4%). Almost half were

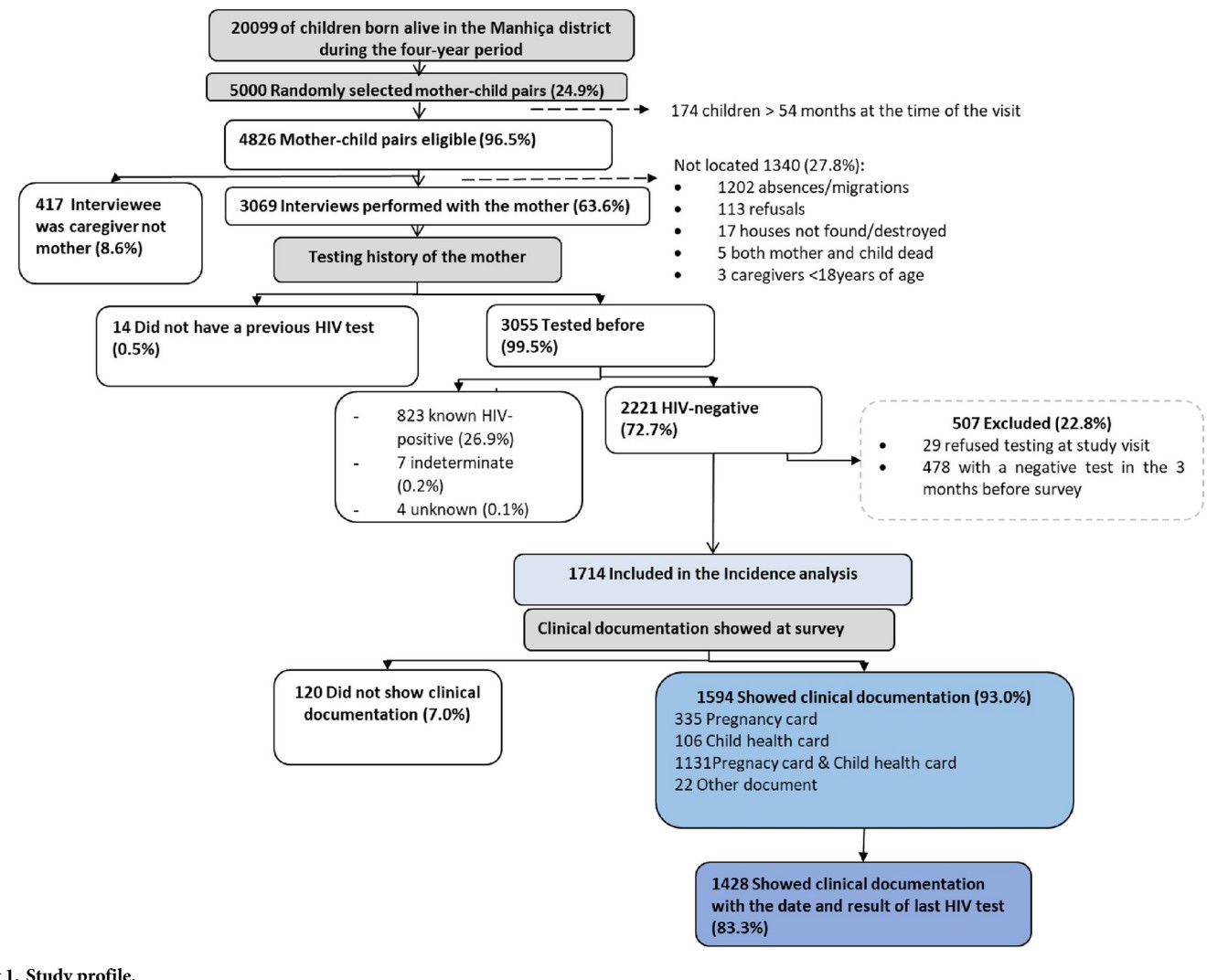

**Fig 1. Study profile.**

**Table 1. Clinical documentation about HIV-testing history showed by the mother at survey.**

| Clinical documentation showed by the mother at survey (N = 1714) | | N | % |
|---|---|---|---|
| Clinical documentation showed at survey | | | |
| Clinical documentation showed at survey | No | 120 | 7.0% |
| | Yes | 1594 | 93.0% |
| Type of clinical document showed | Prenatal card | 335 | 19.5% |
| | Child health card | 106 | 6.2% |
| | Prenatal card & Child health card | 1131 | 66.0% |
| | Other document | 22 | 1.3% |
| | No documentation | 120 | 7.0% |
| Clinical documentation with the date and the result of the last HIV test | No | 166 | 9.7% |
| | Yes | 1428 | 83.3% |
| | No documentation | 120 | 7.0% |

**Table 2. Sociodemografic characteristics of the mother-child pairs included in this analysis at the time of the survey visit.**

| Sociodemografic characteristics of the mother-child pairs included in this analysis at the time of the survey visit (N = 1714) | | N | % |
|---|---|---|---|
| MOTHER | | | |
| Age of the mother at survey in years (IQR) | | 24.8 (20.7–31.5) | |
| Marital status | Single | 211 | 12.3% |
| | Married | 1365 | 79.6% |
| | Divorced/Widowed | 138 | 8.1% |
| Highest educational level | None | 230 | 13.4% |
| | Primary | 1223 | 71.4% |
| | Secondary or higher | 260 | 15.2% |
| | Unknown | 1 | 0.1% |
| Parity | Primipara | 580 | 33.8% |
| | Multipara | 1133 | 66.1% |
| | Unknown | 1 | 0.1% |
| Antenatal clinic visit (any visit) | No | 144 | 8.4% |
| | Yes | 1570 | 91.6% |
| CHILD | | | |
| Age at survey in months (IQR) | | 23.4 (14.5–35.3) | |
| Gender | Female | 822 | 48.0% |
| | Male | 892 | 52.0% |
| Born in Mozambique | No | 28 | 1.6% |
| | Yes | 1686 | 98.4% |
| Breastfeeding at any time | No | 5 | 0.3% |
| | Yes | 1709 | 99.7% |
| Median time of breastfeeding at survey in months (IQR) | | 16.4 (12.0–20.0) | |

For continuous variables, median and interquartile ranges (IQR) were used. For categorical variables, frequencies (counts and proportions) were employed.

multiparous (66.1%) and 8.4% had not attended any antenatal visits during their last pregnancy. Most mothers had breastfed their child at any time (99.7%) and the median time of breastfeeding at survey was 16.4 (IQR: 12.0–20.0) months. The median age of children at time of the survey was 23.4 months (IQR: 14.5–35.3).

## Estimated HIV incidence

Among 1714 mothers included, 57 new HIV infections were detected over 2530.27 person-years of follow-up. Thus, the estimated overall HIV incidence was 2.25 (95% CI: 1.74–2.92) per 100 person-years (Fig 2). There were no differences in HIV incidence by age or education level. Divorced/widowed women had a higher incidence compared to married women, 5.56 (95% CI: 3.16–9.80) and 1.87 (95% CI: 1.36–2.57), respectively.

When the infection date was considered as the date of the survey (no imputation of the date of infection), the estimated overall HIV incidence was 2.21 (95% CI: 1.70–2.86) per 100 person-years, without differences by age or education level. Also, divorced/widowed women had a higher incidence compared to married women, 5.28 (95% CI: 3.00–9.29) and 1.84 (95% CI: 1.34–2.52), respectively (sensitivity analysis 1).

When women with a negative test within the previous three months were included, the estimated overall HIV incidence was 2.04 (95% CI: 1.57–2.65) per 100 person-years (sensitivity

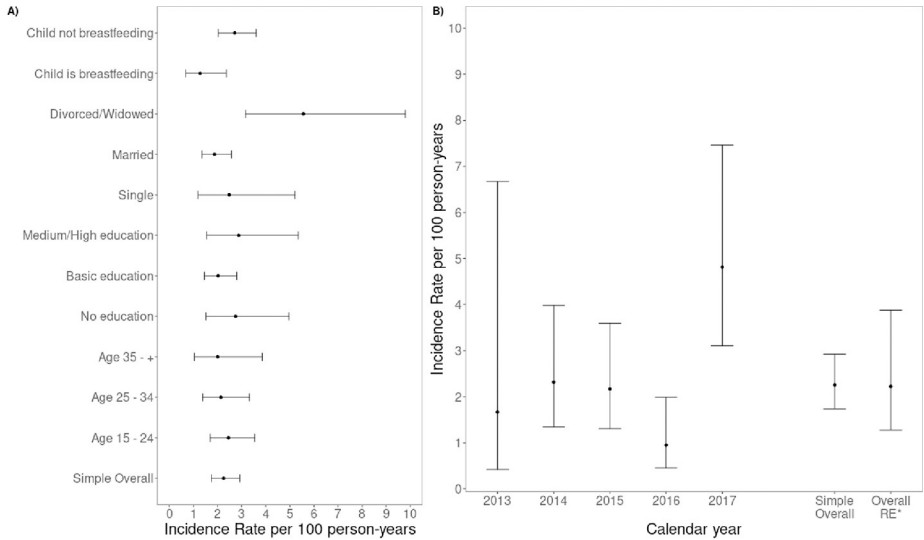

| | Number of new HIV diagnosis | Person-years of follow-up | HIV incidence (per 100 person-year) | 95%CI |
|---|---|---|---|---|
| **Overall** | 57 | 2530.27 | 2.25 | 1.74 - 2.92 |
| **By age (years)** | | | | |
| 15-24 | 28 | 1143.40 | 2.45 | 1.69 - 3.55 |
| 25-34 | 20 | 933.38 | 2.14 | 1.38 - 3.32 |
| >=35 | 9 | 449.02 | 2.00 | 1.04 - 3.85 |
| **By highest educational level** | | | | |
| None | 11 | 400.08 | 2.75 | 1.52 - 4.97 |
| Primary | 36 | 1781.07 | 2.02 | 1.46 - 2.80 |
| Secondary or higher | 10 | 347.59 | 2.88 | 1.55 - 5.35 |
| **By marital status** | | | | |
| Single | 7 | 281.67 | 2.49 | 1.18 - 5.21 |
| Married | 38 | 2032.93 | 1.87 | 1.36 - 2.57 |
| Divorced/Widowed | 12 | 215.67 | 5.56 | 3.16 - 9.80 |
| **By breastfeeding at survey** | | | | |
| Child still breastfeeding | 10 | 786.43 | 1.27 | 0.68 - 2.36 |
| Child stopped breastfeeding | 47 | 1734.85 | 2.71 | 2.04 - 3.61 |
| **By Year of last HIV test** | | | | |
| 2013 | 2 | 119.85 | 1.67 | 0.42 - 6.67 |
| 2014 | 13 | 561.99 | 2.31 | 1.34 - 3.98 |
| 2015 | 15 | 692.03 | 2.17 | 1.31 - 3.59 |
| 2016 | 7 | 739.40 | 0.95 | 0.45 - 1.99 |
| 2017 | 20 | 415.50 | 4.81 | 3.10 - 7.46 |

**Fig 2. Overall HIV incidence and HIV incidence by age, highest educational level, marital status, breastfeeding at survey and year of last HIV test.** The incidence rate of new HIV diagnosis was computed as the division of total positives HIV tests by the total person-time contributed since the most recent negative test. For those women testing HIV positive at the survey, the date of seroconversion was established as a random date between the last negative test and the visit date. Then from this estimated random date, the person-time contribution was computed. For the 95% confidence interval, we used Poisson distribution approximation. The overall is the result of total events divided by the person-time. The overall random-effects (RE) is based on yearly random-intercepts Poisson regression to account for the year- to- year variability. A. Overall HIV incidence and HIV incidence by age, educational level, marital status and breastfeeding at survey. B. HIV incidence by year of last HIV test.

analysis 2) without differences by age or education level. Divorced/widowed women had higher incidence compared to married women, 5.00 (95% CI: 2.84–8.81) and 1.69 (95% CI: 1.22–2.33), respectively.

Of the total number of mothers diagnosed with HIV at the time of the survey, 50.9% (29/57) linked to HIV care at the referral health unit and 42.1% (24/57) started ART within 6 months from the end of the study.

## Discussion

This study evaluated the feasibility of estimating HIV incidence by leveraging a community HIV-testing survey that collected previous HIV-testing history. More than 83% of the women who gave birth during the last four years presented a health card documenting their last HIV test result and date. The HIV test conducted in the household during the survey visit allowed HIV incidence estimation.

This study demonstrated that combining the last documented HIV-test result with a cross-sectional HIV-testing survey is feasible for estimating HIV incidence in a high HIV prevalence population. Previous national population-based surveys in Mozambique found that more than 70% of the children under 23 months had a child health card [23]. Our study showed that mothers also had prenatal cards; thus previous HIV test date and result can be used as a base-line for HIV incidence estimation.

Our results are especially relevant in sub-Saharan African nations committed to the United Nations Secretary-General's Global Strategy for Women's and Children's Health [24], most of which periodically conduct household surveys collecting indicators of women's and children's health [25]. During these surveys, HIV-testing is usually conducted, and participant's health cards are checked to verify health indicators [23, 25], yet the HIV-testing information is seldom collected. Routine transcription of this information would be a simple and low-cost method of improving HIV incidence calculation at sub-national and national levels in low-resource settings while using existing infrastructure. This would increase available evidence of HIV incidence and support evidence-based decision-making [26]. However, ensuring confidentiality and fully informed consent for HIV-testing in all settings, including surveys, is crucial [27].

Despite declines since the introduction of Option B+ and test-and-treat strategies [28], the estimated HIV incidence of 2.25 (95% CI: 1.74–2.92) per 100 person-years among women who gave birth during the last four years remains high among women of reproductive age in southern Mozambique. A prospective study enrolling HIV-negative sexually active women between June 2010 and October 2012 in southern Mozambique estimated an HIV incidence of 4.6 (95% CI: 2.7–7.3) per 100 person-years of follow-up [28]. Another prospective study found an overall incidence of 6.66 per 100 person-years (95% CI: 4.34–10.22) among sexually active women in Maputo province between 2007–2009 [29]. Our cross-sectional survey provided plausible estimates of HIV incidence among women of reproductive age in areas with a high HIV burden when compared with results obtained in longitudinal prospective studies. However, the number of new HIV infections was high given the relatively small sample size and additional studies should validate this method among different HIV prevalence settings. This incidence estimate reinforces the need for ongoing robust investment in programs to prevent new HIV infections in reproductive age women.

Prevalence of HIV infection among pregnant women between 15 to 24 years old attending antenatal clinics has been used as a proxy to estimate trends in HIV incidence, especially in countries with generalised epidemics [30, 31]. This method assumes that prevalence in this age group reflects recent infection [30]. However, several studies have pointed to the change in the

pattern of sexual debut and long term survivals of mother to child transmission as factors affecting the reliability of the use of this method as a proxy of recent HIV incidence in adolescents and young adults [30, 32, 33]. In contrast, our proposed method could be used for HIV surveillance in this age group without need for long, costly prospective studies. Other methods for estimating HIV incidence include statistical modeling approaches such as back-calculation, that uses epidemiological surveillance data to reconstruct HIV infections over time [34]. However, the accuracy of this method is influenced by the distribution of the progression rate as well as the testing rate [34]. Using biomarkers from biological samples collected in cross-sectional studies to identify recent HIV infections is an alternative to estimate incidence, but it is prone to false-recent results in populations with a high level of advanced HIV disease if conducted without viral load and CD4 or ART measurements [35].

We did not find differences in HIV incidence by age, but our study included a narrow age distribution of women in childbearing years, making it difficult to detect statistically and epidemiologically significant differences. In age-representative samples, younger age has been found to be a risk factor of HIV acquisition among women and girls [36]. Similar to previous studies in sub-Saharan Africa [37, 38], our study showed higher HIV incidence among divorced or widowed compared to married women, although these two groups were among the lesser sampled. A Ugandan cohort study between 1999–2011 suggested lower number of sex partners among married women may protect them from new HIV infections [37] and similar protection may be present in this population. Unmarried women of reproductive age, particularly those who are divorced or widowed compromise a vulnerable group that would benefit for specific HIV prevention and testing strategies.

We observed that linking to a health facility for HIV care and treatment is still a challenge for people diagnosed in the community, even with facilitated linkage. Our linkage rates are slightly higher compared to 43.7% [95%CI: 40.8 to 46.6] linkage following community-based testing in our setting between 2014 and 2015 [39]. Other strategies such as community-based organizations have improved community linkage to care in other countries of sub-Saharan Africa [40].

This study has several limitations. First, the sample size for HIV incidence estimation was not defined prior to the survey as the main objective was to estimate prevalence. Second, from the initial sample, 22.8% of women with a previous HIV-negative test were excluded from the analysis because they were not tested at the time of the survey due to refusal or receiving a negative HIV result in previous three months. However, a sensitivity analysis assuming these women as negative showed similar results in HIV incidence and similar distribution for age categories, suggesting this would not affect our results. Third, 120 of the 1714 (7.0%) mothers included in the incidence analysis reported a previous negative HIV test but did not show documentation and 16 of them (13.3%) had a positive HIV test at survey. Data from Manhiça in 2015 showed that more than one third of individuals testing HIV-positive did not disclose their previous positive HIV diagnosis in the health facility (Fuente-Soro et al JIAS 2018). Therefore, we cannot assure that these 16 were really newly acquired infections resulting in an overestimation of the HIV incidence. However, the sensitivity analysis excluding the 120 mothers who did not show documentation did not significantly change the overall and categorical results of HIV incidence. Forth, the date of seroconversion was not known. However, we had a relativey short period of follow-up and the seroconversion date was established as a random-point date between the last negative and the first positive result, a validated methodology [18, 19]. Furthermore, sensitivity analysis without imputation of the seroconversion date showed similar results in HIV incidence and similar distribution for the age categories. Finally, neither women who had not given birth recently nor men were included, as the main objective was to estimate prevalence among HIV-exposed children after B+ implementation; thus we can not generalize our findings to other groups.

## Conclusion

Combining the last-documented HIV test result with a cross-sectional HIV survey is an acheivable strategy to estimate HIV incidence. The routine transcription of the previous HIV-testing information from health cards into national surveys about women and children's health could make HIV incidence monitoring a reality for many Low Middle Income Country (LMIC) settings that struggle to generate periodic HIV incidence data among recently-pregnant women. Validation and cost-effectiveness assessments in prospective cohorts in different HIV prevalence settings would lead to expansion of this approach, after which it could rapidly be implemented in LMIC settings to monitor trends in HIV incidence and inform health policy.

## Acknowledgments

We acknowledge support from the Spanish Ministry of Science, Innovation and Universities through the "Centro de Excelencia Severo Ochoa 2019–2023" Program (CEX2018-000806-S), and support from the Generalitat de Catalunya through the CERCA Program. We want specially acknowledge Elisabeth Salvo for their contributions to this work. The authors gratefully acknowledge the Ministry of Health of Mozambique, our evaluation team, collaborators, and especially all communities and participants involved.

## Author Contributions

**Conceptualization:** Orvalho Augusto, Elisa Lopez-Varela, Marilena Urso, Denise Naniche.

**Formal analysis:** Orvalho Augusto, Sheila Fernández-Luis, Adelino Jose Chingore Juga.

**Funding acquisition:** Marilena Urso.

**Investigation:** Sheila Fernández-Luis, Laura Fuente-Soro.

**Methodology:** Orvalho Augusto, Denise Naniche.

**Project administration:** Laura Fuente-Soro.

**Supervision:** Sheila Fernández-Luis, Laura Fuente-Soro, Tacilta Nhampossa, Denise Naniche.

**Writing – original draft:** Sheila Fernández-Luis.

**Writing – review & editing:** Orvalho Augusto, Laura Fuente-Soro, Tacilta Nhampossa, Elisa Lopez-Varela, Ariel Nhacolo, Edson Bernardo, Helga Guambe, Kwalila Tibana, Jessica Greenberg Cowan, Marilena Urso, Denise Naniche.

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
