## [Decision Letter · Decision Letter 0]

14 Nov 2022

PGPH-D-22-01587

Using testing history to estimate HIV incidence in mothers living in resource-limited settings: Maximizing efficiency of a community health survey in Mozambique.

Dear Sheila Fernández-Luis,

Thank you for submitting your manuscript to PLOS Global Public Health. After careful consideration, we feel that it has merit but does not fully meet PLOS Global Public Health’s publication criteria as it currently stands. Therefore, we invite you to submit a revised version of the manuscript that addresses the points raised during the review process.

Please submit your revised manuscript. If you will need more time than this to complete your revisions, please reply to this message or contact the journal office at globalpubhealth@plos.org. Please include the following items when submitting your revised manuscript:

We look forward to receiving your revised manuscript.

Kind regards,

M Abdullah Yusuf

Academic Editor

Journal Requirements:

2. Please send a completed 'Competing Interests' statement, including any COIs declared by your co-authors. If you have no competing interests to declare, please state "The authors have declared that no competing interests exist". Otherwise please declare all competing interests beginning with the statement "I have read the journal's policy and the authors of this manuscript have the following competing interests:"

3. Please provide a/amend your detailed Financial Disclosure statement. This is published with the article. It must therefore be completed in full sentences and contain the exact wording you wish to be published.

4. Please indicate by return email the full and correct funding information for your study and confirm the order in which funding contributions should appear.

5. Please provide separate figure files in .tif or .eps format only and ensure that all files are under our size limit of 10MB.

Additional Editor Comments (if provided):

I have made some of suggestions which would help to upgrade paper quality. I have attached the file with this assignment. I have suggest some of correction between abstract and method section mainly. Some of analysis you may need to more clarify.

Reviewers' comments:

Reviewer's Responses to Questions

**Comments to the Author**

1. Does this manuscript meet PLOS Global Public Health’s publication criteria? Is the manuscript technically sound, and do the data support the conclusions? The manuscript must describe methodologically and ethically rigorous research with conclusions that are appropriately drawn based on the data presented.

Reviewer #1: Yes

Reviewer #2: Yes

2. Has the statistical analysis been performed appropriately and rigorously?

Reviewer #1: Yes

Reviewer #2: Yes

3. Have the authors made all data underlying the findings in their manuscript fully available (please refer to the Data Availability Statement at the start of the manuscript PDF file)?

Reviewer #1: No

Reviewer #2: Yes

4. Is the manuscript presented in an intelligible fashion and written in standard English?

Reviewer #1: Yes

Reviewer #2: Yes

5. Review Comments to the Author

Reviewer #1: The authors estimated the incidence of HIV among women who recently delivered in southern Mozambique using a cross-sectional household HIV-testing survey leveraging routine health card data on HIV testing to identify incident cases. Following are some minor comments to improve the clarity of the paper:

1. Abstract: “Most mothers (83.3%,1428/1714) had a documented HIV test result and date.” –Please clarify if the remaining 16.7% of the mothers had not been tested for HIV previously as HIV testing was offered to mothers with no prior HIV testing history too according to the Methods in the abstract.

2. Introduction, lines 23-26: It is not clear if and how the current study used data from the rural Health Demographic Surveillance System. The abstract mentions a cross-sectional household HIV-testing survey.

3. Methods, Study design:

- Please clarify if the “broader cross-sectional household survey” was the HDSS.

- Before applying a simple random sampling to identify 5000 children born alive, the first step is to identify and list all children born alive within the four years before study implementation. How many children in total were born alive during the four-year period?

- Why were 5000 children identified, that is, how was the sample size calculated?

- Were there any household with more than one mother with a child born within the past four years? How many mothers per household were selected?

- Was any HIV status determined by verbal autopsy alone in absence of documentation/health card? If yes, how reliable is that?

- How many home visit attempts were made for the HIV test survey?

4. Methods, Study procedures: Do the HIV rapid diagnostic tests from the two manufacturers have same diagnostic accuracy?

5. Methods, Statistical analysis:

- For the second sensitivity analysis, women with a negative test within previous three months were excluded from the analysis. However, participants who “had tested HIV-negative more than three months before the study visit” were offered testing (lines 59-60) suggesting participants with a negative HIV test within last three months were not offered testing. Why? Please clarify.

- A sensitivity analysis excluding mothers without documented prior HIV test (prenatal card or child’s health card), and mothers without history of any prior HIV test or mothers without any antenatal visits during their last pregnancy will be useful.

6. Results:

− What proportion of the 2221 HIV-negative results were undocumented self-reported? How reliable is self-reporting? Is there any previous study on the validity of self-report of HIV testing?

- The section ‘Clinical documentation showed at survey’ needs to be placed before the ‘Sociodemographic characteristics of mother-child pairs at survey’.

Reviewer #2: I have made some of suggestions which wold help to upgrade paper quality. I have attached the file with this assignment. I have suggest some of correction between abstract and method section mainly. Some of analysis you may need to more clarify.

6. PLOS authors have the option to publish the peer review history of their article (what does this mean?). If published, this will include your full peer review and any attached files.

**Do you want your identity to be public for this peer review?** For information about this choice, including consent withdrawal, please see our Privacy Policy.

Reviewer #1: No

Reviewer #2: No

---

## [Editor Report · Decision Letter 1]

8 Mar 2023

PGPH-D-22-01587R1

Using testing history to estimate HIV incidence in mothers living in resource-limited settings: Maximizing efficiency of a community health survey in Mozambique.

Dear Dr. Sheila Fernández-Luis,

Thank you for submitting your manuscript to PLOS Global Public Health. After careful consideration, we feel that it has merit but does not fully meet PLOS Global Public Health’s publication criteria as it currently stands. Therefore, we invite you to submit a revised version of the manuscript that addresses the points raised during the review process.

Please do the needful correction

We look forward to receiving your revised manuscript.

Kind regards,

M Abdullah Yusuf

Academic Editor

Journal Requirements:

Additional Editor Comments (if provided):

Need some revision of the manuscript
---

## [Editor Report · Decision Letter 2]

25 Apr 2023

Using testing history to estimate HIV incidence in mothers living in resource-limited settings: Maximizing efficiency of a community health survey in Mozambique.

PGPH-D-22-01587R2

Dear Fernández-Luis,

We are pleased to inform you that your manuscript 'Using testing history to estimate HIV incidence in mothers living in resource-limited settings: Maximizing efficiency of a community health survey in Mozambique.' has been provisionally accepted for publication in PLOS Global Public Health.

Best regards,

Kévin Jean

Academic Editor